# LEARNING ROBUST MODELS BY COUNTERING SPURIOUS CORRELATIONS

## ABSTRACT

Machine learning has demonstrated remarkable prediction accuracy over $i.i.d$ data, but the accuracy often drops when tested with data from another distribution. One reason behind this accuracy drop is the reliance of models on the features that are only associated with the label in the training distribution, but not the test distribution. This problem is usually known as spurious correlation, confounding factors, or dataset bias. In this paper, we formally study the generalization error bound for this setup with the knowledge of how the spurious features are associated with the label. We also compare our analysis to the widely-accepted domain adaptation error bound and show that our bound can be tighter, with more assumptions that we consider realistic. Further, our analysis naturally offers a set of solutions for this problem, linked to established solutions in various topics about robustness in general, and these solutions all require some understandings of how the spurious features are associated with the label. Finally, we also briefly discuss a method that does not require such an understanding.

## 1 INTRODUCTION

Machine learning, especially deep neural networks, has demonstrated remarkable empirical successes over various benchmarks. One promising next step is to extend such empirical achievements beyond $i.i.d$ benchmarks. If we train a model with data from one distribution (*i.e.*, the source distribution), how can we guarantee the error to be small over other unseen, but related distributions (*i.e.*, target distributions). Quantifying the generalization error over two arbitrary distributions is not useful, thus, we require the distributions of study similar but different: being similar in the sense that there exists a common function that can achieve zero error over both distributions, while being different in the sense that there exists another different function that can only achieve zero error over the training distribution, but not the test distribution.

This problem may not be trivial because the empirical risk minimizer (ERM) may lead the model to learn this second function, a topic studied under different terminologies such as spurious correlations (Vigen, 2015), confounding factors (McDonald, 2014) or dataset bias (Torralba & Efros, 2011). As a result, small empirical error may not mean the model learns what we expect (Geirhos et al., 2019; Wang et al., 2020), thus the model may not be able to perform consistently over other related data.

In particular, our view of the challenges in this topic is illustrated with a toy example in Figure 1 where the model is trained on the source domain data to classify triangle vs. circle and tested on the target domain data. However, the color coincides with the shape on the source domain, so the model may learn either the desired function (relying on shape) or the spurious function (relying on color). The spurious function will not classify the target domain data correctly while the desired function can, but the ERM cannot differentiate them. As one may expect, whether shape or color is considered as desired or spurious is subjective dependent on the task or the data, and in general irrelevant to the statistical nature of the problem. Therefore, our error bound will require the knowledge of the spurious function. While this is a toy example, this scenario surly exists in real world tasks (*e.g.*, Jo & Bengio, 2017; Geirhos et al., 2019; Wang et al., 2020). The contributions of this paper are:

- We analyze the cross-distribution generalization error bound of a model when the model is trained with a distribution with spuriously correlated features, which is formalized as the main theorem of this paper.

Figure 1: A toy example of the main problem focused in this paper.

- We compare our bound to the widely-accepted domain adaptation one (Ben-David et al., 2010) and show that our bound can be tighter under assumptions that we consider realistic.
- Our main theorem naturally offers principled solutions of this problem, and the solutions are linked to many previous established methods for robustness in a broader context.
- As the principled solutions all require some knowledge of the task or the data, our main theorem also leads to a new heuristic absent of the knowledge. This new method may be on a par with the principled solutions, and can outperform the vanilla training empirically.

## 2 RELATED WORK

There is a rich history of learning robust models. We first discuss works in three topics, all centering around the concept of invariance, where invariance intuitively means the model's prediction preserves under certain shift of the data. We then highlight works related to our theoretical discussion.

**Cross-domain Generalization** This line of works probably originates from *domain adaptation* (Ben-David et al., 2007), which studies the problem of training a model over one distribution and test it over another one. Since (Ganin et al., 2016), recent advances along this topic mainly center around the concept of invariance: most techniques leverage different regularizers to learn representations that are invariant to the marginals of these two distributions (*e.g.*, Ghifary et al., 2016; Rozantsev et al., 2018). Further, the community aims beyond the situation that a trained model from domain adaptation may only be applicable to one distribution, and focuses on *domain generalization* (Muandet et al., 2013), which studies the problem of training a model over a collection of distributions and test it with distributions unseen during training. Similarly, most recent methods aim to learn representations invariant to the marginals of the training distributions (*e.g.*, Motiian et al., 2017; Li et al., 2018; Carlucci et al., 2018). Recently, the community extends the study to *domain generalization without domain IDs* to address the real-world situations that domain IDs are unavailable (Wang et al., 2019b), which again focuses on learning representations invariant to specifically designed functions.

**Adversarially Robust Models** The study of robustness against adversarial examples was popularized by the empirical observations that small perturbations on image data can significantly alter the model's prediction (Szegedy et al., 2013; Goodfellow et al., 2015). This observation initiated a line of works building models invariant to such small perturbations (the rigorous definitions of "small perturbations" will not be discussed in details here) (*e.g.*, Lee et al., 2017; Akhtar et al., 2018) and adversarial training (Madry et al., 2018) is currently the most widely-accepted method in terms of empirical defense. On the other hand, the community also aims to develop methods that are provably robust to predefined perturbations (*e.g.*, Wong & Kolter, 2018; Croce & Hein, 2020), which links back to the works of distributional robust models (*e.g.*, Abadeh et al., 2015; Sagawa* et al., 2020), whose central goal is to train models invariant to a predefined shift of distributions. Recent evidence shows key challenges of learning adversarially robust models are spuriously correlated features (Ilyas et al., 2019; Wang et al., 2020), connecting adversarial robustness to the next topic.

**Countering Spurious Correlation** Works along this line usually connects the robustness of a model to its ability of ignoring the spurious correlation in the data, which was also studied under the terminologies of *confounding factors*, or *dataset bias*. With different concrete definitions of the spurious correlation, methods have been developed for various applications, such as image/video classification (*e.g.*, Goyal et al., 2017; Wang et al., 2019a;b; Bahng et al., 2019; Shi et al., 2020), text classification (*e.g.*, He et al., 2019; Clark et al., 2019; Bras et al., 2020; Zhou & Bansal, 2020;

Ko et al., 2020), medical diagnosis (*e.g.*, Zech et al., 2018; Chaibub Neto et al., 2019; Larrazabal et al., 2020) *etc*. The key concept is, as expected, to be invariant to the spurious correlated features.

**Related Theoretical Discussion** Out of a rich collection of theoretical discussions in learning robust models, we only focus on the ones for *unsupervised domain adaptation*, as they will be related to our discussions in the sequel. Popularized by (Ben-David et al., 2007; 2010), these analyses, although in various forms (Mansour et al., 2009; Germain et al., 2016; Zhang et al., 2019; Dhouib et al., 2020), mostly involve two additional terms than standard machine learning generalization bound: one term describes the "learnable" nature of the problem and one term quantifies the differences of the two distributions. This second term probably inspired most of the empirical methods forcing invariant representations from distributions. However, the value of invariance is recently challenged (Wu et al., 2019; Zhao et al., 2019). For example, Zhao et al. (2019) argued that "invariance is not sufficient" by showing counter examples violating the "learnable" nature of the problem and formalized the understanding as that the two distributions have possibly different labeling functions.

**Key Difference:** However, we find the argument of disparity in labeling functions less intuitive, because human will nonetheless be able to agree on the label of an object whichever distribution the object lies in: in the context of this paper, we argue a shared labeling function always exists (in any task reasonable to human), but the ERM model may not have the incentive to learn this function and learns a spurious one instead. As in Figure 1, we formalize the problem as learning against spurious functions, and argue that the central problem is still invariance, but instead of invariance to marginals, we urge for invariance to the spurious function. Our discussion also applies to more than unsupervised domain adaptation and relates to most of the topics discussed in this section.

## 3 GENERALIZATION UNDERSTANDING WITH SPURIOUS CORRELATION

### 3.1 NOTATIONS & BACKGROUND

We consider a binary classification problem from feature space $\mathcal{X} \in \mathbb{R}^p$ to label space $\mathcal{Y} \in \{0, 1\}$. The distribution over $\mathcal{X}$ is denoted as $\mathbf{P}$. A *labeling function* $f : \mathcal{X} \to \mathcal{Y}$ is a function that maps feature $\mathbf{x}$ to its label $\mathbf{y}$. A *hypothesis* or *model* $\theta : \mathcal{X} \to \mathcal{Y}$ is also a function that maps feature to the label. The difference in naming is only because we want to differentiate whether the function is a natural property of the space or distribution (thus called a labeling function) or a function to estimate (thus called a hypothesis or model). The hypothesis space is denoted as $\Theta$. This work concerns with the generalization error across two distributions, namely source and target distribution, denoted as $\mathbf{P}_s$ and $\mathbf{P}_t$ respectively. As stated previously, we are only interested when these two distributions are similar but different: being similar means there exists a *desired labeling function*, $f_d$, that maps any $\mathbf{x} \in \mathcal{X}$ to its label (thus the label $\mathbf{y} := f_d(\mathbf{x})$); being different means there exists a *spurious labeling function*, $f_p$, that for any $\mathbf{x} \sim \mathbf{P}_s$, $f_p(\mathbf{x}) = f_d(\mathbf{x})$. This "similar but different" property will be reiterated as an assumption (**A2**) later. We use $(\mathbf{x}, \mathbf{y})$ to denote a sample, and use $(\mathbf{X}, \mathbf{Y})_{\mathbf{P}}$ to denote a finite dataset if the features are drawn from $\mathbf{P}$. We use $\epsilon_{\mathbf{P}}(\theta)$ to denote the expected risk of $\theta$ over distribution $\mathbf{P}$, and use $\widehat{\cdot}$ to denote estimated term $\cdot$ (*e.g.*, the empirical risk is $\widehat{\epsilon}_{\mathbf{P}}(\widehat{\theta})$). We use $l(\cdot, \cdot)$ to denote a generic loss function. For a dataset $(\mathbf{X}, \mathbf{Y})_{\mathbf{P}}$, if we train a model

$$\widehat{\theta} = \arg\min_{\theta \in \Theta} \sum_{(\mathbf{x}, \mathbf{y}) \in (\mathbf{X}, \mathbf{Y})_{\mathbf{P}}} l(\theta(\mathbf{x}), \mathbf{y}), \tag{1}$$

previous generalization study suggests that we can expect the error rate to be bounded as

$$\epsilon_{\mathbf{P}}(\widehat{\theta}) \leq \widehat{\epsilon}_{\mathbf{P}}(\widehat{\theta}) + \phi(|\Theta|, n, \delta), \tag{2}$$

where $\epsilon_{\mathbf{P}}(\widehat{\theta})$ and $\widehat{\epsilon}_{\mathbf{P}}(\widehat{\theta})$ respectively are

$$\epsilon_{\mathbf{P}}(\widehat{\theta}) = \mathbb{E}_{\mathbf{x} \sim \mathbf{P}} |\widehat{\theta}(\mathbf{x}) - \mathbf{y}| = \mathbb{E}_{\mathbf{x} \sim \mathbf{P}} |\widehat{\theta}(\mathbf{x}) - f_d(\mathbf{x})| \quad \text{and} \quad \widehat{\epsilon}_{\mathbf{P}}(\widehat{\theta}) = \frac{1}{n} \sum_{(\mathbf{x}, \mathbf{y}) \in (\mathbf{X}, \mathbf{Y})_{\mathbf{P}}} |\widehat{\theta}(\mathbf{x}) - \mathbf{y}|,$$

and $\phi(|\Theta|, n, \delta)$ is a function of hypothesis space $|\Theta|$, number of samples $n$, and $\delta$ accounts for the probability when the bound holds. This paper only concerns with this generic form that can subsume many discussions, each with its own assumptions. We refer to these assumptions as **A1**.

**A1**: basic assumptions needed to derived (2), formalized with two examples in appendix A.1.

### 3.2 GENERALIZATION UNDERSTANDING WITH SPURIOUS CORRELATION

Our interest lies in more than the expected performance over samples from the same distribution, but over a different distribution that shares the same labeling function. As we argue previously, the key difficulty in learning a robust model is the existence of the extra labeling function for features sampled from $\mathbf{P}_s$. More formally, we have our first assumption describing this problem

> **A2**: **Existence of Spurious Correlation:** For any $\mathbf{x} \in \mathcal{X}$, $\mathbf{y} := f_d(\mathbf{x})$. We also have a $f_p$ that is different from $f_d$, and for $\mathbf{x} \sim \mathbf{P}_s$, $f_d(\mathbf{x}) = f_p(\mathbf{x})$.

Thus, $\theta$ that learns either $f_d$ or $f_p$ will lead to small source error, but only $\theta$ that learns $f_d$ will lead to small target error. Note that $f_p$ may not exist for an arbitrary $\mathbf{P}_s$. In other words, **A2** can be interpreted as a regulation to $\mathbf{P}_s$ so that $f_p$, while being different from $f_d$, exists for any $\mathbf{x} \sim \mathbf{P}_s$.

In this problem, $f_p$ and $f_d$ are not the same despite $f_p(\mathbf{x}) = f_d(\mathbf{x})$ for any $\mathbf{x} \sim \mathbf{P}_s$, and we consider the differences lie in the features they use. To describe this difference, we introduce the notation $\mathcal{A}(\cdot, \cdot)$, which denotes a set parametrized by the labeling function and the sample, to describe the *active set* of features that are used by the labeling function. By *active set*, we refer to the minimum set of features that a labeling function requires to map a sample to its label. Formally, we define

$$\mathcal{A}(f, \mathbf{x}) = \underset{\mathbf{z} \in \mathcal{X}, f(\mathbf{z}) = f(\mathbf{x})}{\arg \min} |\alpha_{\mathbf{x}}(\mathbf{z})|, \tag{3}$$

where $\alpha_{\mathbf{x}}(\mathbf{z}) = \{i | \mathbf{z}_i = \mathbf{x}_i\}$ is the set of indices by which $\mathbf{z}$ and $\mathbf{x}$ are the same, and $| \cdot |$ measures the cardinality. Although $f_p(\mathbf{x}) = f_d(\mathbf{x})$, $\mathcal{A}(f_p, \mathbf{x})$ and $\mathcal{A}(f_d, \mathbf{x})$ can be different.

Further, we define a new function difference given a sample as

$$d_{\mathbf{x}}(\theta, f) = \max_{\mathbf{z} \in \mathcal{X} : \mathbf{z}_{\mathcal{A}(f, \mathbf{x})} = \mathbf{x}_{\mathcal{A}(f, \mathbf{x})}} |\theta(\mathbf{z}) - f(\mathbf{z})|, \tag{4}$$

where $\mathbf{x}_{\mathcal{A}(f, \mathbf{x})}$ denotes the features of $\mathbf{x}$ indexed by $\mathcal{A}(f, \mathbf{x})$. In other words, the distance describes: given a sample $\mathbf{x}$, what is the maximum disagreement of the two functions $\theta$ and $f$ for all the other data $\mathbf{z} \in \mathcal{X}$ with a constraint that the features indexed by $\mathcal{A}(f, \mathbf{x})$ are the same as those of $\mathbf{x}$. Notice that this difference is not symmetric, as the active set is determined by the second function. By definition, we have $d_{\mathbf{x}}(\theta, f) \geq |\theta(\mathbf{x}) - f(\mathbf{x})|$.

We introduce two more assumptions:

> **A3**: **Separable Labeling Functions:** For any $\mathbf{x} \in \mathcal{X}$, $\mathcal{A}(f_d, \mathbf{x}) \cap \mathcal{A}(f_p, \mathbf{x}) = \emptyset$
>
> **A4**: **Realized Hypothesis:** Given a large enough hypothesis space $\Theta$, for any sample $(\mathbf{x}, \mathbf{y})$, for any $\theta \in \Theta$, which is not a constant mapping, if $\theta(\mathbf{x}) = \mathbf{y}$, then $d_{\mathbf{x}}(\theta, f_d) d_{\mathbf{x}}(\theta, f_p) = 0$

Intuitively, **A3** assumes the active sets of the two labeling functions do not overlap. **A4** assumes a $\theta$ at least learns one labeling function for the sample $\mathbf{x}$ if $\theta$ can map the $\mathbf{x}$ correctly.

Finally, we use the term

$$r(\theta, \mathcal{A}(f, \mathbf{x})) = \max_{\mathbf{x}_{\mathcal{A}(f, \mathbf{x})} \in \mathcal{X}_{\mathcal{A}(f, \mathbf{x})}} |\theta(\mathbf{x}) - \mathbf{y}| \tag{5}$$

to describe how $\theta$ depends on the active set of $f$. Notice that $r(\theta, \mathcal{A}(f, \mathbf{x})) = 1$ alone does not mean $\theta$ depends on the active set of $f$, it only means so when we also have $\theta(\mathbf{x}) = \mathbf{y}$ (see formal discussion in Lemma B.1). With all above, we formalize a new generalization bound as follows:

**Theorem 3.1** (The Curse of Universal Approximation). *With Assumptions **A1-A4**, with probability as least $1 - \delta$, we have*

$$\epsilon_{\mathbf{P}_t}(\theta) \leq \widehat{\epsilon}_{\mathbf{P}_s}(\theta) + c(\theta) + \phi(|\Theta|, n, \delta) \tag{6}$$

*where $c(\theta) = \dfrac{1}{n} \sum_{(\mathbf{x}, \mathbf{y}) \in (\mathbf{X}, \mathbf{Y})_{\mathbf{P}_s}} \mathbb{I}[\theta(\mathbf{x}) = \mathbf{y}] r(\theta, \mathcal{A}(f_p, \mathbf{x}))$.*

$\mathbb{I}[\cdot]$ is a function that returns 1 if the condition $\cdot$ holds true, and 0 otherwise. As $\theta$ may learn $f_p$, $\widehat{\epsilon}_{\mathbf{P}_s}(\theta)$ can no longer alone indicate $\epsilon_{\mathbf{P}_t}(\theta)$, thus we introduce $c(\theta)$ to account for the discrepancy. Intuitively, $c(\theta)$ quantifies the samples that are correctly predicted, but only because the $\theta$ learns $f_p$ for that sample. $c(\theta)$ critically depends on the knowledge of $f_p$.

We name Theorem 3.1 *the curse of universal approximation* to highlight the fact that the existence of $f_p$ is not always obvious, but the models can usually learn it nonetheless. For example, Ilyas et al. (2019) suggest the root to the performance drop over adversarial examples are spurious features, and Wang et al. (2020) demonstrate the existence of human-imperceptible high-frequency spurious signals in image datasets, which may explain several generalization issues of the models. In other words, even in a well-curated dataset that does not seemingly have spurious correlated features, modern machine learning models may still use some spurious features not understood by human, leading to non-robust behaviors when tested over other datasets that human consider similar. This argument may also align with recent discussions suggesting that reducing the model complexity can improve cross-domain generalization (Chuang et al., 2020).

### 3.3    IN COMPARISON TO THE VIEW OF DOMAIN ADAPTATION

We further compare Theorem 3.1 with established understandings of domain adaptation. We summarize the several domain adaptation understandings in the following form (see details in §2):

$$\epsilon_{\mathbf{P}_t}(\theta) \leq \widehat{\epsilon}_{\mathbf{P}_s}(\theta) + D_{\Theta}(\mathbf{P}_s, \mathbf{P}_t) + \lambda + \phi'(|\Theta|, n, \delta) \tag{7}$$

where $D_{\Theta}(\mathbf{P}_s, \mathbf{P}_t)$ quantifies the differences of the two distributions, and $\lambda$ describes the nature of the problem, and usually involves non-estimable terms about the problem or the distributions.

For example, Ben-David et al. (2010) formalize the difference as $\mathcal{H}$-divergence, and describe the corresponding empirical term as ($\Theta\Delta\Theta$ is the set of disagreement between two hypotheses in $\Theta$):

$$D_{\Theta}(\mathbf{P}_s, \mathbf{P}_t) = 1 - \min_{\theta \in \Theta\Delta\Theta} \left( \frac{1}{n} \sum_{\mathbf{x}:\theta(\mathbf{x})=0} \mathbb{I}[\mathbf{x} \in (\mathbf{X}, \mathbf{Y})_{\mathbf{P}_s}] + \frac{1}{n} \sum_{\mathbf{x}:\theta(\mathbf{x})=1} \mathbb{I}[\mathbf{x} \in (\mathbf{X}, \mathbf{Y})_{\mathbf{P}_t}] \right), \tag{8}$$

where $m$ denotes the number of unlabelled samples in $\mathbf{P}_s$ and $\mathbf{P}_t$ each. $\lambda = \epsilon_{\mathbf{P}_t}(\theta^\star) + \epsilon_{\mathbf{P}_s}(\theta^\star)$, where $\theta^\star = \arg\min_{\theta \in \Theta} \epsilon_{\mathbf{P}_t}(\theta) + \epsilon_{\mathbf{P}_s}(\theta)$,

In our formalization, as we assume the $f_d$ applies to any $\mathbf{x} \in \mathcal{X}$ (according to **A2**), $\lambda = 0$ as long as the hypothesis space is large enough. Therefore, the difference mainly lies in the comparison between $c(\theta)$ and $D_{\Theta}(\mathbf{P}_s, \mathbf{P}_t)$. To compare them, we need an extra assumption:

> **A5**: **Sufficiency of Training Samples** for the two finite datasets in the study, *i.e.*, $(\mathbf{X}, \mathbf{Y})_{\mathbf{P}_s}$ and $(\mathbf{X}, \mathbf{Y})_{\mathbf{P}_t}$, for any $\mathbf{x} \in (\mathbf{X}, \mathbf{Y})_{\mathbf{P}_t}$, there exists one or many $\mathbf{z} \in (\mathbf{X}, \mathbf{Y})_{\mathbf{P}_s}$ such that
> $$\mathbf{x} \in \{\mathbf{x}'|\mathbf{x}' \in \mathcal{X} \text{ and } \mathbf{x}'_{\mathcal{A}(f_d, \mathbf{z})} = \mathbf{z}_{\mathcal{A}(f_d, \mathbf{z})}\} \tag{9}$$

**A5** intuitively means the finite training dataset needs to be diverse enough to describe the concept that needs to be learned. For example, imagine building a classifier to classify mammals *vs.* fishes from the distribution of photos to the distribution of sketches, we cannot expect the classifier to do anything good on dolphins if dolphins only appear in the test sketch dataset. **A5** intuitively regulates that if dolphins will appear in the test sketch dataset, they must also appear in the training dataset.

Now, in comparison to (Ben-David et al., 2010), we have

**Theorem 3.2.** *With Assumptions A2-A5, and if $1 - f_d \in \Theta$, we have*

$$c(\theta) \leq D_{\Theta}(\mathbf{P}_s, \mathbf{P}_t) + \frac{1}{n} \sum_{(\mathbf{x},\mathbf{y}) \in (\mathbf{X}, \mathbf{Y})_{\mathbf{P}_t}} \mathbb{I}[\theta(\mathbf{x}) = \mathbf{y}] r(\theta, \mathcal{A}(f_p, \mathbf{x})) \tag{10}$$

*where $c(\theta) = \frac{1}{n} \sum_{(\mathbf{x},\mathbf{y}) \in (\mathbf{X}, \mathbf{Y})_{\mathbf{P}_s}} \mathbb{I}[\theta(\mathbf{x}) = \mathbf{y}] r(\theta, \mathcal{A}(f_p, \mathbf{x}))$ and $D_{\Theta}(\mathbf{P}_s, \mathbf{P}_t)$ is defined as in* (8).

The comparison involves an extra term, $q(\theta) := \frac{1}{n} \sum_{(\mathbf{x},\mathbf{y}) \in (\mathbf{X}, \mathbf{Y})_{\mathbf{P}_t}} \mathbb{I}[\theta(\mathbf{x}) = \mathbf{y}] r(\theta, \mathcal{A}(f_p, \mathbf{x}))$, which intuitively means that if $\theta$ learns $f_p$, how many samples $\theta$ can coincidentally predict correctly over the finite target set used to estimate $D_{\Theta}(\mathbf{P}_s, \mathbf{P}_t)$. For sanity check, if we replace $(\mathbf{X}, \mathbf{Y})_{\mathbf{P}_t}$ with $(\mathbf{X}, \mathbf{Y})_{\mathbf{P}_s}$, $D_{\Theta}(\mathbf{P}_s, \mathbf{P}_t)$ will be evaluated at 0 as it cannot differentiate two identical datasets, and $q(\theta)$ will be the same as $c(\theta)$. On the other hand, if no samples from $(\mathbf{X}, \mathbf{Y})_{\mathbf{P}_t}$ can be mapped correctly with $f_p$ (coincidentally), $q(\theta) = 0$ and $c(\theta)$ will be a lower bound of $D_{\Theta}(\mathbf{P}_s, \mathbf{P}_t)$.

The value of Theorem 3.2 lies in the fact that for an arbitrary target dataset $(\mathbf{X}, \mathbf{Y})_{\mathbf{P}_t}$, no samples out of which can be predicted correctly by learning $f_p$ (a situation likely to occur for arbitrary datasets), $c(\theta)$ will always be a lower bound of $D_{\Theta}(\mathbf{P}_s, \mathbf{P}_t)$.

Further, when Assumption **A5** does not hold, we are unable to derive a clear relationship between $c(\theta)$ and $D_\Theta(\mathbf{P}_s, \mathbf{P}_t)$. The difference is mainly raised as a matter of fact that, intuitively, we are only interested in the problems that are "solvable" (**A5**, *i.e.*, hypothesis that used to reduce the test error in target distribution can be learned from the finite training samples) but "hard to solve" (**A2**, *i.e.*, another labeling function, namely $f_p$, exists for features sampled from the source distribution only), while $D_\Theta(\mathbf{P}_s, \mathbf{P}_t)$ estimates the divergence of two arbitrary distributions.

**Estimation of** $c(\theta)$**.** Finally, due to the limitation of space, we discuss the estimation of $c(\theta)$ in appendix A.2. In short, in practice, while we do not know $f_p$, we usually know $\mathcal{A}(f_p, \mathbf{x})$ through intuition or common sense, such as texture or background of images. Thus, the estimation is to test the whether the model switches its correct prediction when these features are perturbed over the possible space. Also, the search can be terminated once $r(\theta, \mathcal{A}(f_p, \mathbf{x}))$ is evaluated as 1. As one may be aware of, this process is widely known as adversarial attack (*e.g.*, Goodfellow et al., 2015).

# 4 METHODS TO LEARN ROBUST MODELS

In this section, we will take advantage of our analysis to introduce methods that can be used to learn robust models by countering the spurious correlation. First, we discuss the principled solutions that can lead to the error bound in Theorem 3.1 to be smaller. These methods are interestingly linked to many previously established methods in robustness in general. Second, as all above methods will require some knowledge of $f_p$ or $\mathcal{A}(f_p, \mathbf{x})$, we also explore a new method that does not require so.

## 4.1 PRINCIPLED SOLUTIONS OF LEARNING ROBUST MODELS

According to Theorem 3.1, a key to learning robust model is to reduce $c(\theta)$. However, for the convenience during training, we can consider its upper bound

$$c(\theta) \le \frac{1}{n} \sum_{(\mathbf{x}, \mathbf{y}) \in (\mathbf{X}, \mathbf{Y})} r(\theta, \mathcal{A}(f_p, \mathbf{x})) = \frac{1}{n} \sum_{(\mathbf{x}, \mathbf{y}) \in (\mathbf{X}, \mathbf{Y})} \max_{\mathbf{x}_{\mathcal{A}(f_p, \mathbf{x})} \in \mathcal{X}_{\mathcal{A}(f_p, \mathbf{x})}} |\theta(\mathbf{x}) - \mathbf{y}|, \tag{11}$$

which intuitively means that instead of $c(\theta)$ that concerns with the correct prediction based on only $f_p$, now we study any prediction based on only $f_p$.

Also, as we introduced in "estimation of $c(\theta)$": although we barely know $f_p$ in practice, we usually directly know $\mathcal{A}(f_p, \mathbf{x})$ through the intuition or common sense of the data or the task.

**Adversarially robust models (worst-case data augmentation)** The most direct approach of learning robust models will be optimizing to reduce (11) in addition to the generic loss (*i.e.*, $l(\theta(\mathbf{x}), \mathbf{y})$) of a model. Further, as $|\theta(\mathbf{x}) - \mathbf{y}| \le \max_{\mathbf{x}_{\mathcal{A}(f_p, \mathbf{x})} \in \mathcal{X}_{\mathcal{A}(f_p, \mathbf{x})}} |\theta(\mathbf{x}) - \mathbf{y}|$, we can drop the generic loss term, and directly train a model with

$$\min_\theta \frac{1}{n} \sum_{(\mathbf{x}, \mathbf{y}) \in (\mathbf{X}, \mathbf{Y})} \max_{\mathbf{x}_{\mathcal{A}(f_p, \mathbf{x})} \in \mathcal{X}_{\mathcal{A}(f_p, \mathbf{x})}} l(\theta(\mathbf{x}), \mathbf{y}), \tag{12}$$

which is one of the most widely used methods in adversarial robust literature: the adversarial training (Madry et al., 2018), as well as worst-case data augmentation (Fawzi et al., 2016).

**Data augmentation** Alternatively, we can assume

$$\mathbb{E}_{\mathbf{x}_{\mathcal{A}(f_p, \mathbf{x})} \sim \mathbf{P}^{\text{all}}_{\mathcal{A}(f_p, \mathbf{x})}} |\theta(\mathbf{x}) - \mathbf{y}| = \max_{\mathbf{x}_{\mathcal{A}(f_p, \mathbf{x})} \in \mathcal{X}_{\mathcal{A}(f_p, \mathbf{x})}} |\theta(\mathbf{x}) - \mathbf{y}|, \tag{13}$$

where $\mathbf{P}^{\text{all}}$ denotes the distribution that can help to remove the correlation between $f_p$ related features and $\mathbf{y}$ (*e.g.*, a uniform distribution over $\mathcal{X}$ is sufficient, but $\mathbf{P}_s$ will not be good enough). The main strategy is to train with the samples whose $f_p$ related features are randomized, so that the model is expected to learn to ignore the pattern. As $\mathbf{x}$ can usually be sampled by $\mathbf{P}^{\text{all}}$, one can drop the generic loss and train a model with

$$\min_\theta \frac{1}{n} \sum_{(\mathbf{x}, \mathbf{y}) \in (\mathbf{X}, \mathbf{Y})} \mathbb{E}_{\mathbf{x}_{\mathcal{A}(f_p, \mathbf{x})} \sim \mathbf{P}^{\text{all}}_{\mathcal{A}(f_p, \mathbf{x})}} l(\theta(\mathbf{x}), \mathbf{y}). \tag{14}$$

**Regularizing hypothesis space** We can also consider to find a $\Theta_{\text{regularized}}$ so that

$$|\theta(\mathbf{x}) - \mathbf{y}| = \max_{\mathbf{x}_{\mathcal{A}(f_p, \mathbf{x})} \in \mathcal{X}_{\mathcal{A}(f_p, \mathbf{x})}} |\theta(\mathbf{x}) - \mathbf{y}| \quad \text{for} \quad \theta \in \Theta_{\text{regularized}}, \tag{15}$$

which intuitively means that for any $\theta \in \Theta_{\text{regularized}}$, $\theta$ ignores the information from $\mathcal{A}(f_p, \mathbf{x})$. There is a proliferation of recent developments of this thread, either through intuitive understanding of the task or the data (*e.g.*, Wang et al., 2019a;b; Bahng et al., 2019) or through theoretical understanding for rigorously defined perturbations (*e.g.*, Abadeh et al., 2015; Cranko et al., 2020).

### 4.2 LEARNING ROBUST MODELS WITH MINIMUM SUPERVISION

While our analysis suggests that we cannot have a robust model without the knowledge of $f_b$, we continue to ask that what the best we can do without such knowledge. If we use $\mathcal{F}$ to denote the set $\{f_d, f_p\}$ and use $i$ to index its element, we can have the following upper bound to optimize

$$c(\theta) \leq \frac{1}{n} \sum_{(\mathbf{x}, \mathbf{y}) \in (\mathbf{X}, \mathbf{Y})} r(\theta, \mathcal{A}(f_p, \mathbf{x})) \leq \frac{1}{n} \sum_{(\mathbf{x}, \mathbf{y}) \in (\mathbf{X}, \mathbf{Y})} \sum_i r(\theta, \mathcal{A}(\mathcal{F}_i, \mathbf{x})). \tag{16}$$

By optimizing the RHS of (16), we aim to discourage the learning towards functions that *only* rely on one labelling function for each sample for *any* labelling functions. Intuitively, the method is to encourage the model's usage in *all* possible features (either associated with $f_d$ or $f_p$), thus the model may be more robust to the changes of features when dealing with perturbations of the data.

A model using all the features is not expected to be better than a method that only uses the $f_d$ associated features. However, as all methods in §4.1 require specific knowledge of $f_p$, we argue this is a better practice than vanilla training (1) when there are no side information about either $f_d$ or $f_p$.

In practice, as we do not have the knowledge of $\mathcal{F}$, we can use the estimated $\theta$ from previous iteration as a substitute, also as the searching for $\mathcal{A}(f, \mathbf{x})$ can be computationally expensive, we use the gradient information to guide the selection of the features. In summary, this new method, which we name *minimum supervision (MS)*, has the following three major steps at iteration $t$:

- Use $\theta^{(t-1)}$ as a substitute of either $f_d$ or $f_p$.
- Identify $\mathcal{A}(\theta^{(t-1)}, \mathbf{x})$ with top $\rho$ fraction of features according to the gradient.
- Sample $\mathbf{x}_{\mathcal{A}(\theta^{(t-1)}, \mathbf{x})}$ over $\mathcal{X}_{\mathcal{A}(\theta^{(t-1)}, \mathbf{x})}$ and continue to train the model.

Thus, the new method has a hyperparameter $\rho$. Due to the limitation of space, we discuss the detailed algorithm and other practical aspects of the method in appendix A.3.

## 5 EXPERIMENTS

The experiments fall into two scenarios: we first use the two binary classification experiments to support Theorem 3.1 and 3.2 (§5.1); we then test how the new method we introduced in comparison to previously developed methods explicitly using the knowledge of $f_p$ (§5.2).

### 5.1 THEORY SUPPORTING EXPERIMENT

**Synthetic Data with Spurious Correlation** We also Figure 1 to data with $p$ features, where $p/2$ features are related to $f_d$, $p/4$ features are related to $f_p$, and the rest $p/4$ features are independent on labels. Also, $f_d$ is a non-linear function, and $f_p$ is simpler. We test across multiple choices. Overall, the results suggest (1) minimum supervision works better than the vanilla method; (2) $c(\theta)$ is a tighter estimation of the test error than $D_{\Theta}(\mathbf{P}_s, \mathbf{P}_t)$. Details are in appendix C.1.

**Binary Digit Classification over Transferable Adversarial Examples** We also verify the theoretical discussion through a binary digit classification experiment, where the train and validation set are digits 0 and 1 from MNIST train and validation dataset. To create the test set, we first estimate a model, and perform adversarial attacks over this model for the test samples with five adversarial attack methods (C&W (Carlini & Wagner, 2017), DeepFool (Moosavi-Dezfooli et al., 2016), FGSM

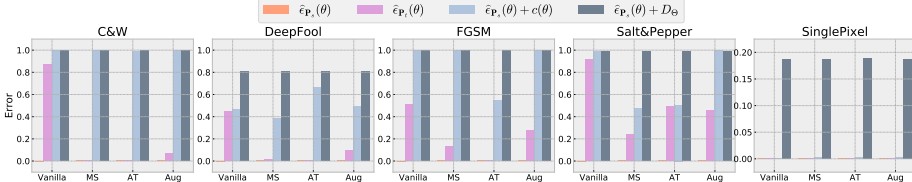

Figure 2: Binary MNIST classification error and estimated bounds. Each panel represents one out-of-domain data generated through an attack method. Four methods are reported in each panel. For each method, four bars are plotted: from left to right, $\widehat{\epsilon}_{\mathbf{P}_s}(\theta)$, $\widehat{\epsilon}_{\mathbf{P}_t}(\theta)$, $\widehat{\epsilon}_{\mathbf{P}_s}(\theta) + c(\theta)$, and $\widehat{\epsilon}_{\mathbf{P}_s}(\theta) + D_\Theta$. Some bars are not visible because the values are small.

| | Vanilla$^\star$ | StylisedIN | LearnedMixin | RUBi | ReBias | MS$^\star$ |
|---|---|---|---|---|---|---|
| Standard Accuracy | 90.8 | 88.4 | 67.9 | 90.5 | 91.9 | **92.1** |
| Weighted Accuracy | 88.8 | 86.6 | 65.9 | 88.6 | 90.5 | **91.3** |
| ImageNet-A | 24.9 | 24.6 | 18.8 | 27.7 | **29.6** | 29.3 |
| ImageNet-Sketch | 41.1 | 40.5 | 36.8 | 42.3 | 41.8 | **42.9** |

Table 1: Results comparison on 9 super-class ImageNet classification. $^\star$Only Vanilla and MS do not leverage any knowledge of the spurious features.

(Goodfellow et al., 2015), Salt&Pepper (Rauber et al., 2017), and SinglePixel (Rauber et al., 2017)). These adversarially generated examples are considered as the test set from another distribution.

An advantage of this setup is that we can have $f_p$ well defined as $1 - f_{adv}$, where the $f_{adv}$ is the function each adversarial attack relies on. Thus, we can assess our analysis on image classification.

We train the models with vanilla method, minimum supervision method (MS), adversarial training (AT), and data augmentation (Aug). In addition to the training error (*i.e.*, $\widehat{\epsilon}_{\mathbf{P}_s}(\theta)$) and test error (*i.e.*, $\widehat{\epsilon}_{\mathbf{P}_t}(\theta)$), we also report $\widehat{\epsilon}_{\mathbf{P}_s}(\theta) + c(\theta)$ and $\widehat{\epsilon}_{\mathbf{P}_s}(\theta) + D_\Theta$. Our results in Figure 2 again align with our analysis: (1) minimum supervision outperforms the vanilla method, but may be inferior to methods using $f_p$ explicitly; (2) $c(\theta)$ is often a tighter estimation of the test error than $D_\Theta(\mathbf{P}_s, \mathbf{P}_t)$.

## 5.2 REAL IMAGE CLASSIFICATION

Finally, we conduct a real-image classification to compare whether our minimum supervision method can be compared to other advanced methods in a more challenging and realistic setting. We follow the setup in (Bahng et al., 2019) and compare the models for a 9 super-class ImageNet classification (Ilyas et al., 2019) with class balanced strategies. Also, we follow (Bahng et al., 2019) to report standard accuracy, weighted accuracy, a scenario where samples with unusual texture are weighted more, and accuracy over ImageNet-A (Hendrycks et al., 2019), a collection of failure cases for most ImageNet trained models. Additionally, we also report the performance over ImageNet-Sketch (Wang et al., 2019a), an independently collected ImageNet test set with only sketch images.

We report the results in Table 1. Our method (MS) outperforms other methods in most situations, which we consider impressive since only MS and vanilla methods are not using any knowledge of $f_p$ or $f_d$. More details of the experiment setup and competing methods are discussed in appendix C.2.

## 6 CONCLUSION

In this paper, we formalized the generalization error when the models can use some spurious features in the training set that are not shared in the test set, a problem widely studied under the terminologies of spurious correlation, confounding factors, or dataset bias. We formalized a new generalization error bound, and compared our bound to the well-established domain adaptation one. More importantly, our theorem naturally offers a set of principled solutions for this problem. These principled solutions are linked to many previous methods for robustness in a broader context. Since all these principled solutions require some knowledge of the spurious correlated features, we also leveraged our theorem to develop a new method that does not require such knowledge.

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

## A    SUPPORTING DISCUSSIONS

### A.1    CONCRETE EXAMPLES OF GENERIC GENERALIZATION BOUND

- when **A1** is "$\Theta$ is finite, $l(\cdot, \cdot)$ is a zero-one loss, samples are *i.i.d*", $\phi(|\Theta|, n, \delta) = \sqrt{(\log(|\Theta|) + \log(1/\delta))/2n}$

- when **A1** is "samples are *i.i.d*", $\phi(|\Theta|, n, \delta) = 2\mathcal{R}(\mathcal{L}) + \sqrt{(\log 1/\delta)/2n}$, where $\mathcal{R}(\mathcal{L})$ stands for Rademacher complexity and $\mathcal{L} = \{l_\theta \mid \theta \in \Theta\}$, where $l_\theta$ is the loss function corresponding to $\theta$.

For more information or more concrete examples of the generic term, one can refer to relevant textbooks such as (Bousquet et al., 2003).

### A.2    ESTIMATION OF $c(\theta)$

The estimation of $c(\theta)$ mainly involves two difficulties: the knowledge of $f_p$ and the computational cost of the search over the entire space $\mathcal{X}$. The first difficulty is usually resolved with intuition or common sense of the data or the task: in practice, we usually directly have the knowledge of $\mathcal{A}(f_p, \mathbf{x})$, *i.e.*, the spuriously correlated features that $f_p$ relies on, such as texture of images. Therefore, the estimation becomes a process to test the whether the model will switch its correct prediction when these features are perturbed over the possible space. The second difficulty can be alleviated due to the fact that the search can be terminated once $r(\theta, \mathcal{A}(f, \mathbf{x}))$ is evaluated as 1. As one may be aware of, this process of searching the entire space with perturbations allowed in a predefined scope to test the model's worst possible prediction for a sample $\mathbf{x}$ is widely known as adversarial attack (Goodfellow et al., 2015). These techniques also usually leverage the knowledge of the model's gradient to accelerate the searching process.

While adversarial attack can offer a fairly accurate estimation of $c(\theta)$, it usually requires heavy computational efforts. As an alternative strategy, many other literature have tested the models with some fixed perturbations of the $\mathbf{x}$, or in other words, taking advantage of the fact that

$$|\theta(\mathbf{x}') - \mathbf{y}| \leq \max_{\mathbf{x}_{\mathcal{A}(f,\mathbf{x})} in \mathcal{X}_{\mathcal{A}(f,\mathbf{x})}} |\theta(\mathbf{x}) - \mathbf{y}| = r(\theta, \mathcal{A}(f, \mathbf{x})), \quad \text{where} \quad \mathbf{x}'_{\mathcal{A}(f,\mathbf{x})} \in \mathcal{X}_{\mathcal{A}(f,\mathbf{x})}. \quad (17)$$

to test a lower bound of $c(\theta)$. There are many works in this thread, and we only list a handful of examples: Jo & Bengio (2017) leveraged Fourier transform to show that models can capture a significant amount of texture information, later Geirhos et al. (2019) showed that CNNs trained with ImageNet are also biased towards texture. With a more concrete definition of the texture, Wang et al. (2020) demonstrated the models can capture high-frequency signals from images, which also links the discussion of learning through bias signals to the adversarial vulnerability issue of models (Ilyas et al., 2019). Similarly, these works mostly depend on a subjective choice of $\mathcal{A}(f_p, \mathbf{x})$, usually given by the knowledge of the data or the task. Although these works did not directly assess $c(\theta)$, $\theta$ usually switched the prediction for sufficient samples to raise an alarm.

### A.3    LEARNING ROBUST MODELS WITH MINIMUM SUPERVISION IN PRACTICE

In practice, as we do not have the knowledge of either $f_d$ or $f_p$ ($\mathcal{F}$), the strategy we use is to estimate the model first and consider our estimated model $\widehat{\theta}$ as a substitute of the labeling function (either $f_d$ or $f_p$). Therefore, at each iteration $t$, we will use the $\widehat{\theta}$ at the previous iteration to identify the active set for the optimization of (16) (in main manuscript).

Further, another question is that when we have $\widehat{\theta}^{t-1}$, how to identify $\mathcal{A}(\widehat{\theta}^{t-1}, \mathbf{x})$, as searching for $\mathcal{A}(\widehat{\theta}^{t-1}, \mathbf{x})$ by the definition can be computationally expensive. Our practical strategy is to use the gradient of $\widehat{\theta}^{t-1}$ to guide the selection of the features. Intuitively, we argue that the the features with larger absolute values of $\partial l(\theta^{t-1}, \mathbf{x}, \mathbf{y})/\partial\theta^{t-1}$ are the features $\widehat{\theta}^{t-1}$ relies on.

Finally, we consider the features with values greater than a threshold $\tau(\rho, \mathbf{g})\}$ are the features that are in $\mathcal{A}(\widehat{\theta}^{t-1}, \mathbf{x})$. The threshold hold is set as the $\rho^{\text{th}}$ quantile of all the calculated gradients for this sample. The algorithm is shown in Algorithm 1

---

**Algorithm 1:** Learning Robust Models with Minimum Supervision

---

**Result:** $\theta^T$

**Input:** $T$, $\rho$, $(\mathbf{X}, \mathbf{Y})$;

initialize $\theta^0$, $t = 1$, $\eta$;

**while** $t \leq T$ **do**

    **for** *sample* $(\mathbf{x}, \mathbf{y})$ **do**

        calculate the gradient $\mathbf{g} = \partial l(\theta^{t-1}, \mathbf{x}, \mathbf{y})/\partial \theta^{t-1}$;

        set the threshold $\tau(\rho, \mathbf{g})$ to be the $\rho^{\text{th}}$ quantile of $|\mathbf{g}|$;

        set $\mathcal{A}(\theta^{t-1}, \mathbf{x}) = \{i||\mathbf{g}_i| \geq \tau(\rho, \mathbf{g})\}$;

        sample $\mathbf{x}'$ where $\mathbf{x}'_{\mathcal{A}(\theta^{t-1}, \mathbf{x})} \in \mathcal{X}_{\mathcal{A}(\theta^{t-1}, \mathbf{x})}$;

        calculate the gradient $\mathbf{g}' = \partial l(\theta^{t-1}, \mathbf{x}', \mathbf{y})/\partial \theta^{t-1}$;

        update the model $\theta^t = \theta^{t-1} - \eta \mathbf{g}'$

    **end**

**end**

---

# B PROOFS OF THEORETICAL DISCUSSIONS

## B.1 LEMMA B.1 AND PROOF

**Lemma B.1.** *With sample $(\mathbf{x}, \mathbf{y})$ and two labeling functions $f_1(\mathbf{x}) = f_2(\mathbf{x}) = \mathbf{y}$, for an estimated $\theta \in \Theta$, if $\theta(\mathbf{x}) = \mathbf{y}$, then with **A3** and **A4**, we have*

$$d_{\mathbf{x}}(\theta, f_1) = 1 \iff r(\theta, \mathcal{A}(f_2, \mathbf{x})) = 1 \tag{18}$$

$\mathbf{x}_{\mathcal{A}(f,\mathbf{x})} \in \mathcal{X}_{\mathcal{A}(f,\mathbf{x})}$ *denotes that the features of $\mathbf{x}$ indexed by $\mathcal{A}(f, \mathbf{x})$ are searched in the entire space.*

*Proof.* If $\theta(\mathbf{x}) = \mathbf{y}$ and $d_{\mathbf{x}}(\theta, f_1) = 1$, according to **A4**, we have $d_{\mathbf{x}}(\theta, f_2) = 0$.

First, we consider one direction $d_{\mathbf{x}}(\theta, f_1) = 1 \implies r(\theta, \mathcal{A}(f_2, \mathbf{x})) = 1$ and we prove this by contradiction.

If the conclusion does not hold, $r(\theta, \mathcal{A}(f_2, \mathbf{x})) = 0$, which means

$$\max_{\mathbf{x}_{\mathcal{A}(f_2,\mathbf{x})} \in \mathcal{X}_{\mathcal{A}(f_2,\mathbf{x})}} |\theta(\mathbf{x}) - \mathbf{y}| = 0 \tag{19}$$

Together with $d_{\mathbf{x}}(\theta, f_2) = 0$, which means

$$\max_{\mathbf{z} \in \mathcal{X}: \mathbf{z}_{\mathcal{A}(f_2,\mathbf{x})} = \mathbf{x}_{\mathcal{A}(f_2,\mathbf{x})}} |\theta(\mathbf{z}) - \mathbf{y}| = 0, \tag{20}$$

we will have

$$\max_{\mathbf{x} \in \mathcal{X}} |\theta(\mathbf{x}) - \mathbf{y}| = 0, \tag{21}$$

which is $\theta(\mathbf{x}) = \mathbf{y}$ for any $\mathbf{x} \in \mathbf{P}$.

This contradicts with the premises in **A4** ($\theta$ is not a constant function).

Second, we consider the other direction $r(\theta, \mathcal{A}(f_2, \mathbf{x})) = 1 \implies d_{\mathbf{x}}(\theta, f_1) = 1$ and we prove this by showing its contrapositive proposition holds. (Its contrapositive proposition is $d_{\mathbf{x}}(\theta, f_1) = 0 \implies r(\theta, \mathcal{A}(f_2, \mathbf{x})) = 0$, because, by definitions, $r$ and $d$ can only be evaluated as 0 or 1).

Because of **A3** ($\mathcal{A}(f_1, \mathbf{x}) \cap \mathcal{A}(f_2, \mathbf{x}) = \emptyset$), we have $d_{\mathbf{x}}(\theta, f_1) \geq r(\theta, \mathcal{A}(f_2, \mathbf{x}))$, thus the contrapositive proposition can be shown trivially. □

## B.2 THEOREM 3.1 AND PROOF

**Theorem.** *With Assumptions **A1**-**A4**, with probability as least $1 - \delta$, we have*

$$\epsilon_{\mathbf{P}_t}(\theta) \leq \widehat{\epsilon}_{\mathbf{P}_s}(\theta) + c(\theta) + \phi(|\Theta|, n, \delta) \tag{22}$$

*where $c(\theta) = \dfrac{1}{n} \sum_{(\mathbf{x},\mathbf{y}) \in (\mathbf{X},\mathbf{Y})_{\mathbf{P}_s}} \mathbb{I}[\theta(\mathbf{x}) = \mathbf{y}] r(\theta, \mathcal{A}(f_p, \mathbf{x})).$*

*Proof.*

$$\widehat{\epsilon}_{\mathbf{P}_s}(\theta) = \frac{1}{n} \sum_{(\mathbf{x},\mathbf{y}) \in (\mathbf{X},\mathbf{Y})_{\mathbf{P}_s}} |\theta(\mathbf{x}) - f(\mathbf{x})| \tag{23}$$

$$= 1 - \frac{1}{n} \sum_{(\mathbf{x},\mathbf{y}) \in (\mathbf{X},\mathbf{Y})_{\mathbf{P}_s}} \left( \mathbb{I}[\theta(\mathbf{x}) = f(\mathbf{x})] \right) \tag{24}$$

$$= 1 - \frac{1}{n} \sum_{(\mathbf{x},\mathbf{y}) \in (\mathbf{X},\mathbf{Y})_{\mathbf{P}_s}} \left( \mathbb{I}[\theta(\mathbf{x}) = f(\mathbf{x})] \mathbb{I}[d_{\mathbf{x}}(\theta, f_d) = 0] + \mathbb{I}[\theta(\mathbf{x}) = f(\mathbf{x})] \mathbb{I}[d_{\mathbf{x}}(\theta, f_d) = 1] \right) \tag{25}$$

$$= 1 - \frac{1}{n} \sum_{(\mathbf{x},\mathbf{y}) \in (\mathbf{X},\mathbf{Y})_{\mathbf{P}_s}} \left( \mathbb{I}[\theta(\mathbf{x}) = f(\mathbf{x})] \mathbb{I}[d_{\mathbf{x}}(\theta, f_d) = 0] \right) - \frac{1}{n} \sum_{(\mathbf{x},\mathbf{y}) \in (\mathbf{X},\mathbf{Y})_{\mathbf{P}_s}} \mathbb{I}[\theta(\mathbf{x}) = f(\mathbf{x})] \mathbb{I}[d_{\mathbf{x}}(\theta, f_d) = 1] \tag{26}$$

$$= \widehat{\epsilon}_d(\theta) - \frac{1}{n} \sum_{(\mathbf{x},\mathbf{y}) \in (\mathbf{X},\mathbf{Y})_{\mathbf{P}_s}} \mathbb{I}[\theta(\mathbf{x}) = f(\mathbf{x})] r(\theta, \mathcal{A}(f_p, \mathbf{x})), \tag{27}$$

where the last line used Lemma B.1.

Thus, we have

$$\widehat{\epsilon}_d(\theta) = \widehat{\epsilon}(\theta) + \frac{1}{n} \sum_{(\mathbf{x},\mathbf{y}) \in (\mathbf{X},\mathbf{Y})_{\mathbf{P}_s}} \mathbb{I}[\theta(\mathbf{x}) = f(\mathbf{x})] r(\theta, \mathcal{A}(f_p, \mathbf{x})) \tag{28}$$

where

$$\widehat{\epsilon}_d(\theta) = 1 - \frac{1}{n} \sum_{(\mathbf{x},\mathbf{y}) \in (\mathbf{X},\mathbf{Y})_{\mathbf{P}_s}} \left( \mathbb{I}[\theta(\mathbf{x}) = f(\mathbf{x})] \mathbb{I}[d_\mathbf{x}(\theta, f_d) = 0] \right), \tag{29}$$

which describes the correctly predicted terms that $\theta$ functions the same as $f_d$ and all the wrongly predicted terms. Therefore, conventional generalization analysis through uniform convergence applies, and we have

$$\epsilon_{\mathbf{P}_t}(\theta) \leq \widehat{\epsilon}_d(\theta) + \phi(|\Theta|, n, \delta) \tag{30}$$

Thus, we have:

$$\epsilon_{\mathbf{P}_t}(\theta) \leq \widehat{\epsilon}_{\mathbf{P}_s}(\theta) + \frac{1}{n} \sum_{(\mathbf{x},\mathbf{y}) \in (\mathbf{X},\mathbf{Y})_{\mathbf{P}_s}} \mathbb{I}[\theta(\mathbf{x}) = \mathbf{y}] r(\theta, \mathcal{A}(f_p, \mathbf{x})) + \phi(|\Theta|, n, \delta) \tag{31}$$

$\square$

### B.3 Theorem 3.2 and Proof

**Theorem.** *With Assumptions A2-A5, and if $1 - f_d \in \Theta$, we have*

$$c(\theta) \leq D_\Theta(\mathbf{P}_s, \mathbf{P}_t) + \frac{1}{n} \sum_{(\mathbf{x},\mathbf{y}) \in (\mathbf{X},\mathbf{Y})_{\mathbf{P}_t}} \mathbb{I}[\theta(\mathbf{x}) = \mathbf{y}] r(\theta, \mathcal{A}(f_p, \mathbf{x})) \tag{32}$$

*where $c(\theta) = \frac{1}{n} \sum_{(\mathbf{x},\mathbf{y}) \in (\mathbf{X},\mathbf{Y})_{\mathbf{P}_s}} \mathbb{I}[\theta(\mathbf{x}) = \mathbf{y}] r(\theta, \mathcal{A}(f_p, \mathbf{x}))$ and $D_\Theta(\mathbf{P}_s, \mathbf{P}_t)$ is defined as in (8).*

*Proof.* By definition, $g(\mathbf{x}) \in \Theta\Delta\Theta \iff g(\mathbf{x}) = \theta(\mathbf{x}) \oplus \theta'(\mathbf{x})$ for some $\theta, \theta' \in \Theta$, together with Lemma 2 and Lemma 3 of (Ben-David et al., 2010), we have

$$D_\Theta(\mathbf{P}_s, \mathbf{P}_t) = \frac{1}{n} \max_{\theta,\theta' \in \Theta} \Big| \sum_{(\mathbf{x},\mathbf{y}) \in (\mathbf{X},\mathbf{Y})_{\mathbf{P}_s}} |\theta(\mathbf{x}) - \theta'(\mathbf{x})| - \sum_{(\mathbf{x},\mathbf{y}) \in (\mathbf{X},\mathbf{Y})_{\mathbf{P}_t}} |\theta(\mathbf{x}) - \theta'(\mathbf{x})| \Big| \tag{33}$$

$$\geq \frac{1}{n} \Big| \sum_{(\mathbf{x},\mathbf{y}) \in (\mathbf{X},\mathbf{Y})_{\mathbf{P}_s}} |\theta(\mathbf{x}) - f_z(\mathbf{x})| - \sum_{(\mathbf{x},\mathbf{y}) \in (\mathbf{X},\mathbf{Y})_{\mathbf{P}_t}} |\theta(\mathbf{x}) - f_z(\mathbf{x})| \Big| \tag{34}$$

$$= \frac{1}{n} \Big| \sum_{(\mathbf{x},\mathbf{y}) \in (\mathbf{X},\mathbf{Y})_{\mathbf{P}_s}} \mathbb{I}[\theta(\mathbf{x}) = \mathbf{y}] - \sum_{(\mathbf{x},\mathbf{y}) \in (\mathbf{X},\mathbf{Y})_{\mathbf{P}_t}} \mathbb{I}[\theta(\mathbf{x}) = \mathbf{y}] \Big| \tag{35}$$

$$= \frac{1}{n} \Big| \sum_{(\mathbf{x},\mathbf{y}) \in (\mathbf{X},\mathbf{Y})_{\mathbf{P}_s}} \mathbb{I}[\theta(\mathbf{x}) = \mathbf{y}] \mathbb{I}[r(\theta, \mathcal{A}(f_p, \mathbf{x})) = 1] - \sum_{(\mathbf{x},\mathbf{y}) \in (\mathbf{X},\mathbf{Y})_{\mathbf{P}_t}} \mathbb{I}[\theta(\mathbf{x}) = \mathbf{y}] \mathbb{I}[r(\theta, \mathcal{A}(f_p, \mathbf{x})) = 1]$$

$$\tag{36}$$

$$+ \sum_{(\mathbf{x},\mathbf{y}) \in (\mathbf{X},\mathbf{Y})_{\mathbf{P}_s}} \mathbb{I}[\theta(\mathbf{x}) = \mathbf{y}] \mathbb{I}[r(\theta, \mathcal{A}(f_p, \mathbf{x})) = 0] - \sum_{(\mathbf{x},\mathbf{y}) \in (\mathbf{X},\mathbf{Y})_{\mathbf{P}_t}} \mathbb{I}[\theta(\mathbf{x}) = \mathbf{y}] \mathbb{I}[r(\theta, \mathcal{A}(f_p, \mathbf{x})) = 0] \Big|$$

$$\tag{37}$$

$$= \frac{1}{n} \Big| \sum_{(\mathbf{x},\mathbf{y}) \in (\mathbf{X},\mathbf{Y})_{\mathbf{P}_s}} \mathbb{I}[\theta(\mathbf{x}) = \mathbf{y}] r(\theta, \mathcal{A}(f_p, \mathbf{x})) - \sum_{(\mathbf{x},\mathbf{y}) \in (\mathbf{X},\mathbf{Y})_{\mathbf{P}_t}} \mathbb{I}[\theta(\mathbf{x}) = \mathbf{y}] r(\theta, \mathcal{A}(f_p, \mathbf{x})) \Big|$$

$$\tag{38}$$

$$\geq c(\theta) - \sum_{(\mathbf{x},\mathbf{y}) \in (\mathbf{X},\mathbf{Y})_{\mathbf{P}_t}} \mathbb{I}[\theta(\mathbf{x}) = \mathbf{y}] r(\theta, \mathcal{A}(f_p, \mathbf{x})) \tag{39}$$

First line: see Lemma 2 and Lemma 3 of (Ben-David et al., 2010).

Second line: if $1 - f_d \in \Theta$, and we use $f_z$ to denote $1 - f_d$.

Fifth line is a result of using that fact that

$$\sum_{(\mathbf{x},\mathbf{y})\in(\mathbf{X},\mathbf{Y})_{\mathbf{P}_s}} \mathbb{I}[\theta(\mathbf{x}) = \mathbf{y}]\mathbb{I}[r(\theta, \mathcal{A}(f_p, \mathbf{x})) = 0] = \sum_{(\mathbf{x},\mathbf{y})\in(\mathbf{X},\mathbf{Y})_{\mathbf{P}_t}} \mathbb{I}[\theta(\mathbf{x}) = \mathbf{y}]\mathbb{I}[r(\theta, \mathcal{A}(f_p, \mathbf{x})) = 0]$$

(40)

as a result of our assumptions. Now we present the details of this argument:

According to **A4**, if $\theta(\mathbf{x}) = \mathbf{y}$, $d_x(\theta, f_d)d_x(\theta, f_p) = 0$. Since $r(\theta, \mathcal{A}(f_p, \mathbf{x})) = 0$, $d_x(\theta, f_p)$ cannot be 0 unless $\theta$ is a constant mapping that maps every sample to 0 (which will contradicts **A4**). Thus, we have $d_x(\theta, f_d) = 0$.

Therefore, we can rewrite the left-hand term following

$$\sum_{(\mathbf{x},\mathbf{y})\in(\mathbf{X},\mathbf{Y})_{\mathbf{P}_s}} \mathbb{I}[\theta(\mathbf{x}) = \mathbf{y}]\mathbb{I}[r(\theta, \mathcal{A}(f_p, \mathbf{x})) = 0] = \sum_{(\mathbf{x},\mathbf{y})\in(\mathbf{X},\mathbf{Y})_{\mathbf{P}_s}} \mathbb{I}[\theta(\mathbf{x}) = \mathbf{y}]\mathbb{I}[d_x(\theta, f_d) = 0]$$

(41)

and similarly

$$\sum_{(\mathbf{x},\mathbf{y})\in(\mathbf{X},\mathbf{Y})_{\mathbf{P}_t}} \mathbb{I}[\theta(\mathbf{x}) = \mathbf{y}]\mathbb{I}[r(\theta, \mathcal{A}(f_p, \mathbf{x})) = 0] = \sum_{(\mathbf{x},\mathbf{y})\in(\mathbf{X},\mathbf{Y})_{\mathbf{P}_t}} \mathbb{I}[\theta(\mathbf{x}) = \mathbf{y}]\mathbb{I}[d_x(\theta, f_d) = 0] \quad (42)$$

We recap the definition of $d_x(\cdot, \cdot)$, thus $d_x(\theta, f_d) = 0$ means

$$d_{\mathbf{x}}(\theta, f_d) = \max_{\mathbf{z}\in\mathcal{X}:\mathbf{z}_{\mathcal{A}(f,\mathbf{x})}=\mathbf{x}_{\mathcal{A}(f_d,\mathbf{x})}} |\theta(\mathbf{z}) - f_d(\mathbf{z})| = 0$$

(43)

Therefore $d_x(\theta, f_d) = 0$ implies $\mathbb{I}(\theta(\mathbf{x}) = \mathbf{y})$, and

$$|\theta(\mathbf{z}) - f_d(\mathbf{z})| = 0 \quad \forall \quad \mathbf{z}_{\mathcal{A}(f_d,\mathbf{x})} = \mathbf{x}_{\mathcal{A}(f_d,\mathbf{x})}$$

(44)

Therefore, we can continue to rewrite the left-hand term following

$$\sum_{(\mathbf{x},\mathbf{y})\in(\mathbf{X},\mathbf{Y})_{\mathbf{P}_s}} \mathbb{I}[\theta(\mathbf{x}) = \mathbf{y}]\mathbb{I}[d_x(\theta, f_d) = 0] = \sum_{(\mathbf{x},\mathbf{y})\in(\mathbf{X},\mathbf{Y})_{\mathbf{P}_s}} \mathbb{I}[\theta(\mathbf{z}) - f_d(\mathbf{z})] = \sum_{(\mathbf{x},\mathbf{y})\in(\mathbf{X},\mathbf{Y})_{\mathbf{P}_s}} \mathbb{I}[\theta(\mathbf{x}) - f_d(\mathbf{x})]$$

(45)

and similarly

$$\sum_{(\mathbf{x},\mathbf{y})\in(\mathbf{X},\mathbf{Y})_{\mathbf{P}_t}} \mathbb{I}[\theta(\mathbf{x}) = \mathbf{y}]\mathbb{I}[d_x(\theta, f_d) = 0] = \sum_{(\mathbf{x},\mathbf{y})\in(\mathbf{X},\mathbf{Y})_{\mathbf{P}_t}} \mathbb{I}[\theta(\mathbf{z}) - f_d(\mathbf{z})]$$

(46)

where $\mathbf{z}$ denotes any $\mathbf{z} \in \mathcal{X}$ and $\mathbf{z}_{\mathcal{A}(f_d,\mathbf{x})} = \mathbf{x}_{\mathcal{A}(f_d,\mathbf{x})}$.

Further, because of **A5**, we have

$$\sum_{(\mathbf{x},\mathbf{y})\in(\mathbf{X},\mathbf{Y})_{\mathbf{P}_t}} \mathbb{I}[\theta(\mathbf{z}) - f_d(\mathbf{z})] = \sum_{(\mathbf{x},\mathbf{y})\in(\mathbf{X},\mathbf{Y})_{\mathbf{P}_s}} \mathbb{I}[\theta(\mathbf{x}) - f_d(\mathbf{x})].$$

(47)

Thus, we showed the (40) holds and conclude our proof.

$\square$

# C   ADDITIONAL EXPERIMENTS

## C.1   THEORETICAL SUPPORTING EXPERIMENTS

**Synthetic Data with Spurious Correlation**   We extend the setup in Figure 1 to generate the synthetic dataset to test our methods. We study a binary classification problem over the data with $n$ samples and $p$ features, denoted as $\mathbf{X} \in \mathcal{R}^{n \times p}$. For every training and validation sample $i$, we generate feature $j$ as following:

$$
\mathbf{X}_j^{(i)} \sim \begin{cases} N(0,1) & \text{if } 1 \le j \le 3p/4 \\ N(1,1) & \text{if } 3p/4 < j \le p, \text{ and } y^{(i)} = 1, \quad \text{w.p. } \rho \\ N(-1,1) & \text{if } 3p/4 < j \le p, \text{ and } y^{(i)} = 0, \quad \text{w.p. } \rho \\ N(0,1) & \text{if } 3p/4 < j \le p, \quad \text{w.p. } 1-\rho \end{cases},
$$

In contrast, testing data are simply sampled with $\mathbf{x}_j^{(i)} \sim N(0,1)$.

To generate the label for training, validation, and test data, we sample two effect size vectors $\beta_1 \in \mathcal{R}^{p/4}$ and $\beta_2 \in \mathcal{R}^{p/4}$ whose each coefficient is sampled from a Normal distribution. We then generate two intermediate variables:

$$
\mathbf{c}_1^{(i)} = \mathbf{X}_{1,2,\dots,p/4}^{(i)} \beta_1 \quad \text{and} \quad \mathbf{c}_2^{(i)} = \mathbf{X}_{1,2,\dots,p/4}^{(i)} \beta_2
$$

Then we transform these continuous intermediate variables into binary intermediate variables via Bernoulli sampling with the outcome of the inverse logit function ($g^{-1}(\cdot)$) over current responses, *i.e.*,

$$
\mathbf{r}_1^{(i)} = \text{Ber}(g^{-1}(\mathbf{c}_1^{(i)})) \quad \text{and} \quad \mathbf{r}_2^{(i)} = \text{Ber}(g^{-1}(\mathbf{c}_2^{(i)}))
$$

Finally, the label for sample $i$ is determined as $\mathbf{y}^{(i)} = \mathbb{I}(\mathbf{r}_1^{(i)} = \mathbf{r}_2^{(i)})$, where $\mathbb{I}$ is the function that returns 1 if the condition holds and 0 otherwise.

Intuitively, we create a dataset of $p$ features, half of the features are generalizable across train, validation and test datasets through a non-linear decision boundary, one-forth of the features are independent of the label, and the remaining features are spuriously correlated features: these features are correlated with the labels in train and validation set, but independent with the label in test dataset. There are about $c\dot{n}$ train and validation samples have the correlated features.

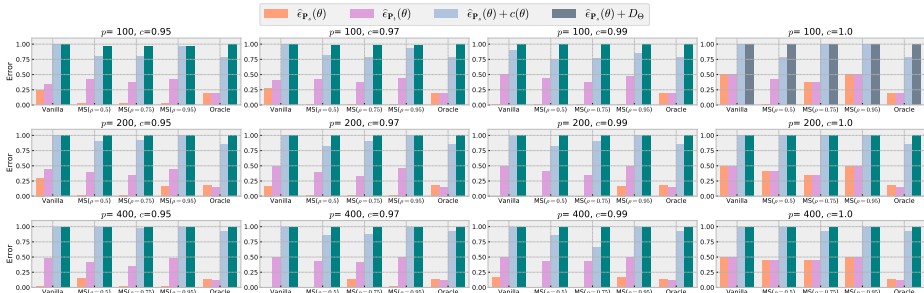

Figure 3: Results of Synthetic Data with Spurious Correlation. Each panel represents one setting. Five methods are reported in each panel. For each method, four bars are plotted: from left to right, $\widehat{\epsilon}_{\mathbf{P}_s}(\theta), \widehat{\epsilon}_{\mathbf{P}_t}(\theta), \widehat{\epsilon}_{\mathbf{P}_s}(\theta) + c(\theta)$, and $\widehat{\epsilon}_{\mathbf{P}_s}(\theta) + D_\Theta$.

Results are reported in Figure 3, where each setup we ran 3 random seeds and report the mean and standard deviation. We train a vanilla method, minimum supervision method with different hyperparamter $\rho$, and an oracle method that uses data augmentation to randomized the previously known spurious features. The results show the advantage of the new method consistently, although still not compared to the method with prior knowledge. We also calculate the $c(\theta)$ as we perform adversarial attacks over the spuriously correlated feature space, we also calculate $D_\Theta$ as defined in (8). We compared $\widehat{\epsilon}_{\mathbf{P}_s}(\theta) + c(\theta)$ and $\widehat{\epsilon}_{\mathbf{P}_s}(\theta) + D_\Theta$ and the results suggest that clearly $c(\theta)$ offers a more accurate assessment of the target error than $D_\Theta$.

### C.2 REAL IMAGE CLASSIFICATION: MORE DETAILS

The main experiment setup follows the setup of (Bahng et al., 2019), and the setup can be conveniently replicated by the GitHub repo associated with the paper (Bahng et al., 2019). Although results of ImageNet-C are also reporeted by (Bahng et al., 2019), their github repo does not provide the corresponding replication scripts, so we also skip the information. Additionally, we report another ImageNet level test set that is independently collected, and has only sketch images.

We rename the "bias" and "unbiased" in (Bahng et al., 2019) to "standard accuracy" and "weighted accuracy" to align the terms we use in this paper and also help to explain the results. Intuitively, "weighted accuracy" refers to the evaluation mechanism that the test samples with unusual texture will have more weights.

Again, following the setup in (Bahng et al., 2019), the base network is ResNet, and we compare with the vanilla network, and several methods that are designed to reduce the texture bias: including StylisedIN (Geirhos et al., 2019), LearnedMixin (Clark et al., 2019), RUBi (Cadene et al., 2019) and ReBias (Bahng et al., 2019).

Finally, to get the reported performance, our MS method uses an extra heuristic, such as we only optimize (16) for half of the batch, and optimize the other batch with the vanilla training (1). Despite this heuristic used, the main message remains: MS method, as a method that does not use the knowledge of the spurious correlated features, can compete with the methods that use the knowledge explicitly.

