# OpenReview forum: "Learning Robust Models by Countering Spurious Correlations"
_ICLR.cc/2021/Conference — Reject_

### Official Review · AnonReviewer3 · 2020-10-20
**This paper aims to derive the generalization error bound for a domain adaptation problem with spuriously correlated features. The derived bound was shown to tighten the standard error bound in domain-adaptation.**

**Rating:** 3
**Confidence:** 4

**Review:**


Quality:
Pros:
- The role of spuriously correlated features in causing poor generalization in the target domain is well described.

Cons:
- The derived bound consider unrealistically simple setup and makes strong assumptions that makes the derived bound practically useless.
- Even if all the assumptions are right the suggested solutions for training a "robust" model is very ad-hoc and unlikely to be robust.

Clarity:
The paper is poorly written and most parts of the paper are difficult to follow.

Originality/Significance:
Although deriving a generalization bound  as a function of spurious features is original, due to strong assumptions, the derived bound is unlikely to have much practical value and  make a significant impact in the field.


Detailed Comments:
This work approaches cross-domain generalization from a relatively easier and somewhat unrealistic direction and does not consider the more challenging setup of emerging classes/subclasses in the target domain nor it discusses the effect of class imbalance on generalization.

It was not clear how invariance to spurious function can be imposed in ERM when the spurious function is not known during training.

A3 is a very strong assumption. Two labeling functions both predicting same labels on the training set yet not sharing even a single feature is not a very practical assumption.

Section 3.2 is hastily written and does not read well.

It is not clear what was so interesting about this finding "This argument may also align with recent discussions suggesting that reducing the model complexity can improve cross-domain generalization (Chuang et al., 2020)."
In the context of structural/empirical risk minimization Vapnik has demonstrated (more than 40 years ago) that model complexity can be reduced by reducing VC dimension, which in turn will tighten the VC bound and improve generalizability on test data. This  result is distribution independent and thus would hold nicely for cross-domain generalization setting  considered in this paper as well (no emerging classes in the test data).

It is not immediately clear what the benefit of the added c(\theta) term would be. Comparing equation (6) with (2) we now have the test error upper bounded by a term that is larger by c(\theta), which makes the bound loose compared to equation (2). This is a standard problem for domain-adaptation. So, I don't consider this as a flaw but some insight would have been useful.

A5 (Sufficient of training samples) is also a very strong assumption that can not be easily justified in real-world domain adaptation problems.  Although theorem 3.2 shows that the extra term c(\theta) is less than or equal to the empirical term in the standard domain adaption error bound, the difference may not be practically significant.

It is not clear under what conditions r(\theta;A(fp; x))=1 suggest an adversarial sample. Possibly only when an adversarial feature is generated by perturbing spurious features that don't overlap with nonspurious features? However, an adversarial sample can also be generated by perturbing only a small fraction of non-spurious features.

"In short, in practice, while we do not know fp, we usually know A(fp; x) through intuition or common sense, such as texture or background of images. Thus, the estimation is to test the whether the model switches its correct prediction when these features are perturbed over the possible space."
For modern machine learning problems that use semantically rich and high-dimensional features (i.e., ResNet) such a task can quickly become impractical. One cannot easily and uniquely identify the subset of features to perturb as there will be an impractically large number of combinations to try.

In section 4.2 the solution that was suggested involves learning only with the top \rho fraction of features ranked according to the gradient. This is somewhat counter-intuitive as this approach will inevitably create sudden spikes in the gradient between different epochs which is expected to make the gradient less useful for feature selection. The loss function being minimized is not the same between different epochs.

---

### Official Review · AnonReviewer1 · 2020-10-25
**Interesting work on robust learning against spurious correlation but some unclear parts need to be clarified**

**Rating:** 5
**Confidence:** 3

**Review:**

This paper studies the problem that spurious features in the training set can cause accuracy drop in the test phase. They formalize a generalization error bound for this setup and provide two solutions, one principled solution with the knowledge of spurious correlated features and one minimal supervision (MS) method without knowing this information, based on their bound. They also have some experimental results demonstrating the effectiveness of their proposed MS method.

Pros:

-The problem of spurious correlations widely exists in real-world machine learning tasks. This paper starts from analyzing the generalization with spurious correlation, comparing with existing results in domain adaptation, and then links to several existing solutions and provides their solution without requiring the spurious features knowledge.

Cons:

-The paper seems to interchangeably use spurious correlation, confounding factors and dataset bias. But my feeling is that they have different definitions in specific problem settings.

In this paper, spurious correlation means there exists another labeling function other than the true mapping function that can achieve zero training error, but cannot generalize well on test data. But the dataset bias may generally mean that the training data (joint) distribution differs from the test data (joint) distribution. So in the latter case, the differences can only lie in labels, for example, noisy training labels. Those differences should be carefully discussed.

-Another concern is that it is hard to verify the correctness of the selected active set defined in Eq 3. In the paper, it is said that those features are usually selected by intuition or common sense. It is still unclear to me how to select them in practice, and how this affects the performance.

-There are some typos in Section 4.2 regarding f_d and f_p, for example, in the first sentence, f_b should be f_p. In my understanding, the proposed method in Eq 16 uses both f_d and f_p associated features, how to guarantee that the model does not learn spurious correlated features only, in the extreme case?

Overall take

-The paper studies an interesting problem and provides some useful analysis, but some unclear parts need to be clarified. And the notations should be carefully checked.

---

### Official Review · AnonReviewer2 · 2020-10-25
**Nice theorem and practical method with some concerns regarding the assumptions**

**Rating:** 6
**Confidence:** 4

**Review:**

Summary:
This paper formalizes a new generalization error bound for the problem of spurious correlation (a.k.a. confounding factors or dataset bias) and shows that it is tighter than the well-established domain adaptation one under realistic assumptions. The analysis leads to a set of solutions linking to established solutions. It further proposes a practical solution that does not require explicit knowledge of spurious correlated features as the established ones need.

################################################

Reasons for score:
In general, I like the flow of the paper. It does a decent job of explaining the proposed new bound, how it is superior to the existing bound, how it is connected to the general formulation and existing methods. The proposed practical method also shows promising results. However, I have some concerns / questions regarding its assumptions. Also, several typos need to be fixed.

################################################

Pros:

+mostly well-written and easy to follow

+the proposed bound is tighter than the existing one under reasonable assumptions

+experiment results support the proposed theoretical result as well as the effectiveness of the proposed method.

+the proposed method works surprisingly well

Cons:

-some assumptions might be a bit strong and not very clear stated

-not negligible number of typos


################################################




1.The x used in the theorems seem to assume the features to be disentangled but this is usually not the case. Features are usually mixed together.

2.The author(s) claim that “in practice, we usually directly have the knowledge of
A(fp; x), i.e., the spuriously correlated features that fp relies on, such as texture of images.” However, this is usually not the case.

3.In A5 the exact equality seems to be a strong assumption.

4.I have a relatively hard time bridging the proposed method and the theorem. In particular, I think most of the time, the model is a combination of both f_p and f_d rather than either f_p or f_d?

################################################

Typos / Formatting issues:
-“Thus, the estimation is to test the whether the model”, extra “the”

-sec 4.2 first line, “f_b”, typo; second line, “what the best”, missing “is”

-sec 5.1, “We also Figure 1 to data with p features”??? I did not understand this sentence.

-Figure 3 is too small and the texts on x-axis are barely readable. Besides, there are five different colors used in the figure while there are only four in the legend.

-several equations in the appendix goes beyond the boundary

-eq17 “in”, typo

################################################

questions:

-Thm3.2 What is the intuition of having the assumption 1-f_s \in \Theta?

-What are the details of the attack the paper uses (e.g. number of steps, epsilon, etc.) for the experiment shown in Figure2? I wonder if the attack becomes stronger, will the performance gap become even larger?

-How do you estimate D_\Theta in practice for the baseline method?

---

### Official Review · AnonReviewer4 · 2020-10-29
**Promising direction, but needs significant further clarity**

**Rating:** 4
**Confidence:** 2

**Review:**

Summary:
The paper considers the problem of spurious features affecting robustness of machine learning models. The paper's goal is to show that if we can make assumptions on the knowledge of the spurious features that may exist in the dataset, we can better analyze the error on potential test sets.

Strengths:
1. The paper provides a good overview of different approaches within the areas of robustness and the "spurious features" phenomena, and the framing makes intuitive sense (albeit restrictive -- the belief that there always exists one labeling function for the source/target domain rules out situations where there is a temporal shift, or there are unobserved variables, yet we would still want a robust model)

2. The experimental results on MS seem promising, given that the method does not use knowledge of the spurious features, although somewhat limiting in terms of the range of experiments/datasets.


Weaknesses:
In general, I unfortunately found the paper very difficult to follow, with Sections 3 and 4 taking so much space that the experiments in Section 5 were very hastily described. One suggestion I would have would be to ground the analysis in an example -- perhaps that in Figure 1 -- to make it easier to discuss the different variables in the analysis without becoming too abstract and confusing.

1. I found the notation in particular quite difficult to follow through, with many terms introduced and explained in a rather rushed way. Particularly, I think section A2 could be  improved, including a more detailed description of d_x(\theta, f) and providing more intuition for r(\theta, A(f,x)). It would also help to break down some of the definitions, such as c(\theta) in (10)  into English descriptions first to help follow the different components.

2. There is little justification / explanation for why theta^(t-1) can be considered a good substitute f_d/ f_p?

3. I found the description of the experiments very limited. For example in the synthetic data experiment, what does it mean that "f_p is simpler"? What do the "multiple choices" refer to?

4. At times, the authors mix up terms such as confounding factors, spurious correlations, spurious features, dataset bias, etc. but there are subtle differences in how the different works the authors cite define these terms. For example, dataset bias can be due to a lot more factors than spurious features learned by the model, while confounding factors can describe variables not included in the data at all, potentially making learning the right model from the data impossible. I think the authors should characterize the types of features they consider a lot more rigorously.

Recommendation:
I recommend rejection. While the ideas and intuition are definitely promising directions, I believe the paper in its current state needs significant improvement with respect to clarity.

Further Comments
- For Figure 2, it might be better to plot 1.0-Error to make it easier to visualize all bars.
- I was a bit confused by the difference between the P_s/P_t and (X,Y)P_s / (X,Y)P_t notations.

Typos:
pg. 1
"we require the distributions of study similar but different" --> "to be similar but different"
"this scenario surly exists in real world tasks" --> surely, it would also be good to describe such examples in the text (and why this is a reasonable assumption to make).
"because human will nonetheless be able to agree" --> humans
pg. 3
"basic assumptions needed to derived (2)" --> derive
pg. 6
"Thus, the estimation is to test the whether the model" --> test whether
pg. 7
"we then test how the new method we introduced in comparison to previously developed methods" --> how the new method we introduced performs in comparison to
"We also Figure 1 to data with p features" --> unclear

---

### Author Response · Authors · 2020-11-24
**Rebuttal**

Dear Reviewers,

Thank you very much for the assessments and constructive comments, we appreciate the recognition of the importance of the problem this paper focuses on. We are also grateful for the constructive comments, especially the requirements for further clarifications of certain points. We will continue to develop the paper to clarify these points.

---

### Decision · Program_Chairs · 2021-01-07
**Final Decision**

**Decision:**

Reject

**Comment:**

Reviewers raised concerns about the paper's clarity (interchangeable use of subtly different terms, notation, typos), and how realistic/practical certain assumptions are. The authors are encouraged to incorporate the reviewers' detailed comments for a future submission.